# Characterization of Isolated Human Astrocytes from Aging Brain

**DOI:** 10.3390/ijms26073416

**Published:** 2025-04-05

**Authors:** Geidy E. Serrano, Sidra Aslam, Jessica E. Walker, Ignazio S. Piras, Matthew J. Huentelman, Richard A. Arce, Michael J. Glass, Anthony J. Intorcia, Katsuko E. Suszczewicz, Claryssa I. Borja, Madison P. Cline, Sanaria H. Qiji, Ileana Lorenzini, Suet Theng Beh, Monica Mariner, Addison Krupp, Rylee McHattie, Anissa Shull, Zekiel R. Wermager, Thomas G. Beach

**Affiliations:** 1Banner Sun Health Research Institute, Sun City, AZ 85351, USA; sidra.aslam@bannerhealth.com (S.A.); zekiel.wermager@bannerhealth.com (Z.R.W.);; 2Translational Genomics Research Institute, Phoenix, AZ 85004, USA; ipiras@tgen.org (I.S.P.); mhuentelman@tgen.org (M.J.H.)

**Keywords:** progressive supranuclear palsy, Alzheimer’s disease, Parkinson’s disease, whole-cell-suspension, neurodegeneration, ubiquitinization, autophagy

## Abstract

Astrocytes have multiple crucial roles, including maintaining brain homeostasis and synaptic function, performing phagocytic clearance, and responding to injury and repair. It has been suggested that astrocyte performance is progressively impaired with aging, leading to imbalances in the brain’s internal milieu that eventually impact neuronal function and lead to neurodegeneration. Until now, most evidence of astrocytic dysfunction in aging has come from experiments done with whole tissue homogenates, astrocytes collected by laser capture, or cell cultures derived from animal models or cell lines. In this study, we used postmortem-derived whole cells sorted with anti-GFAP antibodies to compare the unbiased, whole-transcriptomes of human astrocytes from control, older non-impaired individuals and subjects with different neurodegenerative diseases, such as Parkinson’s disease (PD), Alzheimer’s disease (ADD), and progressive supranuclear palsy (PSP). We found hundreds of dysregulated genes between disease and control astrocytes. In addition, we identified numerous genes shared between these common neurodegenerative disorders that are similarly dysregulated; in particular, UBC a gene for ubiquitin, which is a protein integral to cellular homeostasis and critically important in regulating function and outcomes of proteins under cellular stress, was upregulated in PSP, PD, and ADD when compared to control.

## 1. Introduction

Understanding the molecular events behind the normal physiological functions of astrocytes is crucial to unmask the roles that astrocytes may play in neurological disorders. The functions of astrocytes in the healthy adult brain are considered to involve potassium buffering, interstitial volume control, and maintenance of a low interstitial glutamate concentration. These functions are crucial for brain homeostasis and synaptic function [1,2,3,4,5]. Reactive gliosis, a component of neuroinflammation that involves structural and metabolic changes in astrocytes and microglia, is often a prominent feature of neurological disorders like Parkinson’s disease (PD), progressive supranuclear palsy (PSP), and Alzheimer’s disease dementia (ADD). Therefore, balanced astrocytic phagocytic clearance and response to injury and repair are crucial for neuronal recovery following detrimental events. It has been suggested that astrocyte performance is progressively impaired with aging, leading to imbalances in the brain’s internal environment that eventually impact neuronal function and lead to neurodegeneration [1,4,6,7,8,9].

Nevertheless, until now, the vast majority of evidence related to astrocytic dysfunction in aging and neurodegenerative disorders has come from experiments done with whole tissue homogenates, astrocytes collected by laser capture, or cell cultures derived from animal models or cell lines [8,10,11,12,13]. In this study, we used an enriched population of whole human astrocytes to conduct an unbiased comparison of their gene expression in common neurodegenerative diseases and clinically normal subjects.

## 2. Results

Cases with less than 40 million reads or more than 55% of reads that could not be mapped to the human or animal genome were eliminated. WCDS were enriched for astrocytes; this was confirmed visually with GFAP immunostains of the cell suspension after sorting (Figure 1B) by qPCR, where only cell-specific probes for GFAP and Actin consistently amplified with CT values smaller than 30, and by enrichment plots of fGSEA produced using *plotEnrichmnet* function, which showed only the Astrocyte- specific gene list as enriched compared to all other cell-specific genes (Appendix A).

There were no significant differences in group age means (*p* = 0.92). However, the youngest group (PD) had a mean age of 82, while for the oldest group (CN), the mean age was 89 (Table 1). PMI and RIN were also similar between groups (*p* = 0.17), varying from 2.9 to 4.4 h and from 7.7 to 10.0, respectively. As expected, pathologies typical of ADD, PSP and LB pathology were significantly different between groups, with ADD having the highest densities of amyloid plaques and neurofibrillary tangles, PSP the highest level of pathological tau (summation of neuronal and glia tau scores) and PD with the highest densities of alpha-synuclein pathology (Table 1). Frontal cortex tissue adjacent to the sample collected for whole-cell suspensions stained for GFAP showed an increased number of reactive astrocytes in PSP cases compared to controls, PD, and ADD (Figure 2). Differential expression analysis was used to compare gene expression from each diseased group to CN. We found hundreds of dysregulated genes in each diseased group. PSP cases had the largest number of dysregulated genes compared to CN (1968), while ILBD only had one downregulated gene compared to the CN group that has not yet been characterized by other studies (Table 2). A complete list of DEGs is reported in Appendix A, and all data will be posted in Synapse (Project ID: syn64618348).

### 2.1. PSP Comparisons

When PSP genes were compared to CN, we found a total of 1933 upregulated genes and 35 downregulated genes, while expression of UBC, GFAP, MTURN, GLUL, EEF1A1, MAP1A, and MAP1B were some of the most significant upregulated genes with over 8 log2(FC) > N changes (Figure 3A). When PSP was compared to ADD, we observed a total of 89 genes dysregulated, with 88 of those being upregulated. Some of the most upregulated genes in PSP compared to ADD were GFAP, CDR1, BCYRN1, and DUSP26 (Figure 3B). When PSP was compared to PD, there were a total of 243 genes dysregulated, with 239 of those being downregulated in PSP, particularly NAT8L, FTO, and TUBB2A (Figure 3C).

Pathway enrichment analysis revealed the differential processes associated with astrocytes. DEGs obtained from the PSP and CN cases comparison showed an enrichment of dysregulated genes associated with multiple biological pathways, some of the most important involving synaptic pathways, chemokine signaling, protein modification processes, and proteolysis (Figure 4A), while the DEGs found from the PSP and ADD cases comparison suggest dysregulation of cellular senescence, cytokine-cytokine receptor, Inositol phosphate metabolism, and other cellular processes and interactions (Figure 4B). The PD vs. PSP comparison highlights dysregulated genes associated with axonogenesis, glial cell differentiation, histone modification, mitochondrion organization, neurogenesis, pyrophosphatase activity, and regulation of plasma membrane-bounded cell projection organization (Complete list in Appendix A).

### 2.2. ADD Comparison

When ADD was compared to CN, we found a total of 754 genes dysregulated; 749 of those were upregulated genes, and 5 were downregulated (SBF2-AS1, LOC100506675, ANGPTL7, SLC10A7, SNORD3A). UBC, MTRNR2L8, HSPA1A, and HSPA1B were the most significantly changed genes in ADD in comparison to controls. Interestingly, comparing PD with ADD showed dysregulation of only 4 genes: PAR4, SLC7A5, MIR548AN, and NCL. Thrombin receptor PAR4 was the only gene that was upregulated in ADD when compared to PD; however, the dysregulation was one of the largest observed in our comparisons: 28 log2(FC) > N. Pathway enrichment analysis of the DEGs in ADD highlighted dysregulation of synapses, proteolysis, and signaling pathways (Figure 4C).

### 2.3. PD Comparison

When comparing PD astrocytes to CN, we found a total of 1150 dysregulated genes, with 53 genes showing upregulation (UBC, HSFY1, OR10J3, and EEF1A2) and 1097 downregulation. Multiple microRNAs were highly downregulated, showing over 20 log2(FC) > N changes. Other genes that were highly downregulated included multiple chemokines, multiple immunoglobins, DRD4, and GDF1. Pathway enrichment analysis of DEGs showed dysregulation of translocation of olfactory receptors, synapses, synaptic vesicle cycle, and neurotrophic signaling pathways (Figure 4D). Another important comparison was made between PD and ILBD. This comparison showed that a total of 135 genes were upregulated, and 15 were downregulated in PD when compared to possible prodromal cases (ILBD). Some of the most affected genes were DEAF1, DGKQ, OMA1, and ATG101, all showing an upregulation of 9 log2(FC) > N. Enriched pathways were very similar to those observed in PD vs. CN; however, both comparisons only have 5 dysregulated genes in common: RPL18A, PPDPF, LCE1A, SNORD38B, and RNVU1-17.

### 2.4. All Diseased Cases vs. CN

To better understand if there are astrocyte genes commonly dysregulated in all neurodegenerative disorders used in this study, we compared the list of dysregulated genes for each disease vs. CN comparison. We found that PSP vs. CN and ADD vs. CN share 669 common dysregulated genes, while PD-CN vs. ADD-CN share the least number of dysregulated genes with only 10 genes in common (UBC, SBF2-AS1, RN7SL1, and RN7SL2). Every gene commonly dysregulated (n = 691; comparison #1) was further analyzed for protein–protein interactions (PPI) network using the STRING database. We found 541 nodes and 4952 edges; the overlapping analysis identified five common hub genes (ACTB, EGFR, CALM3, HSP90AB1, and UBC; Figure 5).

Another approach for looking at common genes was to combine all disease cases and compare them all with the non-symptomatic cases (CN and ILBD together) using another differential expression analysis. This comparison resulted in a total of 503 DEGs, 364 upregulated and 139 downregulated (Comparison #2). This suggested again common dysregulation of ubiquitin-related genes, with UBC, UBA2, UAB6, USP9Y, USP25, and USP3 in all diseases. This PPI network includes 268 nodes and 1124 edges, while the overlapping analysis identified five common hub genes (UBC, YBX1, UTY, RPL4, and HSPA1B; Figure 6). Besides the common astrocytic dysregulation of ubiquitin and related genes, we also identified commonly affected biological pathways, such as synaptic dysregulation, transcriptional regulation, VEGFA-VEGFR2 pathway, and chaperone-mediated autophagy, just to mention some of the pathways with genes in common.

Furthermore, we combined the PPIs from our comparison #1 and #2 using a merge tool of Cytoscape software v3.9.1 to find the interconnected and intersected core dysregulated genes and validated the importance of predicted hub genes from each list. UBC, RPL4, and HSPA1A were identified as common hub genes (Figure 7), indicating their central role in the intersected network of dysregulated astrocytic genes across multiple neurodegenerative diseases.

The complete dataset for the study is securely posted at Synapse (Project ID: syn64618348).

## 3. Discussion

Multiple studies seem to identify several possible mechanisms of dysfunction in astrocytes from diseased individuals, and some have suggested that one therapeutic strategy would be to replace dysfunctional astrocytes with transplanted healthy astrocytes [8,14]. This was directly tested by Barres’ group [14] using an amyotrophic lateral sclerosis (ALS) model and indirectly tested by Lee’s group [15] using a PD model. Barres’ group showed that astrocytes from human embryonic stem cells (hESCs) were safely injected into ALS-mice and rat models and seemed to delay disease onset, slow down disease progression, and extend life expectancy. Lee‘s group injected astrocytes as well as neural progenitor cells (NPC) into the midbrain of a PD-rat model and showed that TH+ cells in the striatum were more numerous and exhibited healthier and more mature neuronal maturity than those in the control grafts; treated rats showed dramatic behavioral restoration compared to those just treated with NPC injections or shams [8,15]. Our results support this rationale, as we showed how astrocytes of common neurodegenerative disorders seem to be universally dysregulated when compared to aging normal individuals, suggesting that targeting astrocyte dysfunction could be beneficial for common neurodegenerative diseases.

Our main goal was to study human astrocyte-enriched whole cell suspensions. We are using a unique approach of whole cell analysis rather than individual nuclei analysis for a couple of reasons. We hypothesize that we will be able to analyze a larger number of transcripts by using whole cell compared to nuclei isolation [11,16,17,18,19]. In addition, we believe that by grouping or enriching a population with only one cell type, we should be able to capture smaller changes that might be only present in that type of cell. Whole-homogenate analysis can give completely misleading results, as any biochemical constituent that is selectively localized to the depleted cells will appear to be “down-regulated”, whereas, in fact, it has most likely been lost only as an “innocent bystander”. Also, a relevant loss or increase might be completely missed if the biochemical entity is found in many cell types, diluting the “lost” signal from the cell of interest, especially if that cell type is uncommon or rare.

We analyzed RNA transcript changes in PSP, ADD, and PD human astrocytes when compared to controls. All diseased group comparisons showed prominent dysregulation of genes that are involved in synaptic pathways, metabolic regulation, gene regulation, dephosphorylation, protein modification processes, and inflammation. It is well known that astrocytes interact and are affected by many different cell types influencing inflammatory response, especially microglia [20]. Liddelow et al. 2017 [21] and others have described how heterogenous astrocytes are and how, depending on the type of astrocyte, the reactivity of these cells might have neurotoxic/pro-inflammatory or neuroprotective responses. Our astrocytes-enriched samples included both types of astrocytes, class A and B astrocytes [21], but in this analysis, we found that previously reported genes specific to type A and B astrocytes were not dysregulated in disease when compared to aged controls. Our future direction will be to study this heterogeneity with a different technology that could add information to our results and determine if, at a single cell level, we could identify and study changes of different classes of astrocytes in these diseases.

Perhaps not surprisingly, given the glial tauopathy present in PSP, the most dysregulated astrocyte-enriched populations were those derived from PSP cases. It is well known that astrocytes of PSP cases have aggregated 4-repeat tau that seems to increase their reactivity. In this study, we observed that when compared to controls, 1933 genes were upregulated, and 35 were downregulated. Some of the most upregulated genes were astrocyte marker genes, such as GFAP and GLUL [22,23], but also genes associated with protein phosphorylation regulation, such as microtubule-associated proteins MAP1A and MAP1B, but not MAPT, which was previously reported to be downregulated in PSP astrocytes [3,24,25,26,27]. Most downregulated genes in this comparison were mostly microRNAs or genes not yet studied. Remarkably, GFAP upregulation was also observed in PSP when the group was compared to CN or ADD but not when compared to PD or any of the other groups. This finding was interesting as one would have expected that ADD astrocytes would be more upregulated than PD astrocytes [6,11,23]. We observed a similar trend on the GFAP immunostain (Figure 2), where the controls and ADD showed a non-statistically higher percentage of area occupied by reactive astrocytes. However, one of our study’s limitations is the small number of cases and high comorbidity in each group, including controls, that perhaps prevented us from reaching significance in some of these comparisons.

Other dysregulated genes that are worth mentioning are genes that could explain possible astrocytic dysfunction in ADD, for example, downregulation of SLC10A7 and SNORD3A, which respectively are genes associated with intracellular calcium regulation, and reduced resistance to oxidative and ER stress, as well as upregulations of heat shock HSPA1A, and HSPA1B, proteins associated with apoptosis and ubiquitin-proteasome pathway [18,28,29,30]. When comparing PD with control, there were downregulations of many microRNA [31,32,33] as well as chemokines such as CXC14 and CXC15 and immunoglobins genes. This suggests a suppression of inflammatory response in this group, which has been previously suggested by others [31,32,33,34,35], but is not commonly observed in brain regions with higher LB-associated neurodegeneration, such as substantia nigra [36]. Our data seems to indicate that astrocytes in the frontal cortex of PD subjects, even though affected by LB pathology, do not have the same type of inflammatory response as other brain regions. A possible explanation is offered by F. Giovannoni and F.J. Quintana 2020, who suggest that the proliferation of astrocytes and their reactivity vary depending on the type of injury and degree of damage in each brain region [35].

In this study, we also included ILBD cases, a group of subjects that never had parkinsonism during life, but for whom autopsy confirmed the presence of brain a-synuclein pathology and, hence, represent a probable prodromal stage of PD. Changes observed in this group might suggest astrocytic dysfunction at early stages of a-synuclein pathology. The comparison of ILBD to control only showed the downregulation of one uncharacterized RNA gene. When the groups were compared to PD, we were able to observe the upregulation of OMA1. This mitochondrial enzyme plays a key role in regulating mitochondrial morphology and stress signaling, as well as autophagy-related protein 101 (ATG101), suggesting an early compensatory attempt [2,37,38,39,40]. Another upregulated gene that caught our attention was DGKQ, a diacylglycerol kinase (DGK) that has been proposed as a PD risk gene. DGKs seem to be associated with diverse biological events, such as growth factor/cytokine-dependent cell proliferation and motility, immune responses, and glucose metabolism; multiple recent studies suggest that it could be a good target for therapeutic intervention [41,42,43].

STRING was used to identify the portrait genes with the highest level of interaction with one another as this could provide insight into the synergistic effects of altered expression. This analysis emphasized three main machineries involved in protein degradation that seem to be affected in all neurodegenerative enriched astrocytes: autophagy, calcium signal transduction pathway, and the ubiquitin proteasome system (UPS). Autophagy involves lysosomal degradation, and a large component is regulated by chaperone proteins, while Ca^2+^ is a universal and versatile signaling molecule that controls many cellular processes, including neurotransmission, cell metabolism, cell death, and organelle communication, including endoplasmic reticulum, mitochondria, Golgi complex, and lysosomes [11,44]. Previous studies suggest that depending on the type of ubiquitination, the conjugated protein is subjected to different fates. UPS consists of two key components: the ubiquitination system, which selects and targets proteins toward degradation by ubiquitinating them, and the proteasome. It is well known that ubiquitination-mediated control contributes to the maintenance of cellular homeostasis, and dysregulation of ubiquitination reactions plays a relevant role in the pathogenic states of neurodegenerative diseases [45,46,47]. For years, scientists have noticed the presence of ubiquitin inclusions in multiple intracellular abnormally aggregated proteins that are pathological, such as tau and LB. This appears to be a common feature for many disorders where proteins get abnormally aggregated intracellularly and seems to correlate with the common neurodegeneration observed in many neurological diseases [48,49,50,51]. It has also been described how important the ubiquitin system is in glia cells, but to our knowledge, none have shown dysregulation of this system in astrocytes from common neurodegenerative diseases [52]. Ubiquitination can alter the molecular functions of tagged substrates with respect to protein turnover, biological activity, subcellular localization, or protein–protein interaction. The UPS has been implicated in pathways that regulate neurotransmitter release, synaptic membrane receptor turnover, and synaptic plasticity and, as a result, affects a wide variety of cellular processes, with chronic overexpression potentially inducing synaptic dysfunction in neurons. Our results showed that all these pathways are dysregulated in diseased astrocytes. We believe that these results should be validated in targeted arrays in a larger number of cases and see if we could see similar changes when we analyze these diseases using single-cell preparations. Our results strongly support the rationale that modulation of astrocytes should be further evaluated as a potential therapeutic strategy [53,54]; perhaps maintaining healthy astrocytes and an ubiquitin system would prevent abnormal intracellular protein aggregation that results in neurodegeneration.

## 4. Material and Methods

### 4.1. Subject Selection and Characterization

Fresh brain samples came from subjects who were volunteers in the Arizona Study of Aging and Neurodegenerative Disorders (AZSAND) and the Brain and Body Donation Program (BBDP; www.brainandbodydonationprogram.org), a longitudinal clinicopathological study of healthy aging, cognition, and movement in the elderly since 1997 in Sun City, Arizona [43,44]. All human subjects or their legally authorized representatives consented to participation in the Brain and Body Donation Program through an IRB-approved consent (WCG IRB, Puyallup, WA, USA). The study protocol conforms to the principles of the Declaration of Helsinki. Cases for this study were selected based on their clinicopathological diagnosis, favoring those with the shortest postmortem intervals (PMI). Control subjects were cognitively unimpaired without parkinsonism or dementia and with lower amounts of AD pathology (CN; n = 3). We also included a small number of cases without PD parkinsonism or cognitive impairment but with Lewy bodies in their brain, termed incidental Lewy body disease (ILBD; n = 3). Impaired subjects included cases with PSP (n = 5), ADD (n = 4), and PD (n = 6). A complete neuropathological examination was performed using standard AZSAND methods [43,44]. Neurofibrillary degeneration was staged on the thick frozen sections by the original method of Braak [45,46], and neuropathological ADD diagnoses were made when neuritic plaque densities and Braak stage met “intermediate” or “high” criteria according to NIA-AA criteria [47,48,49,50]. Non-ADD conditions were diagnosed using standard clinicopathological criteria, with international consensus criteria for those disorders where these were available [51,52]. Kruskal–Wallis with Dunn’s Multiple Comparison Test were used to analyze group differences from all demographic and pathological data reported in Table 1 using GraphPad Prism Software (version 10.4).

### 4.2. Whole-Cell-Dissociated-Suspension Preparation

Whole-cell-dissociated-suspensions (WCDS) were generated from the prefrontal cortex of fresh human brain tissue, as previously reported [55]. Briefly, fresh bilateral coronal sections of the frontal lobe were collected just anterior to the genu of the corpus callosum at autopsy. The grey matter was dissected and finely minced and incubated in Accutase (AT104, Innovative Cell Technologies, San Diego, CA, USA) for 4 h at 4 °C, followed by mechanical disruption by repetitive pipetting. Homogenates were centrifuged, and Accutase was replaced by Hank’s balanced salt solution (HBSS), following cell filtration using 100 and 70 µm filters. Myelin, neuropil, and other cellular debris were removed using 30% and 70% Percoll (17-0891-0, GE Healthcare, Chicago, IL, USA). WCDS were stored in a cryoprotectant solution (90% FBA and 10% DMSO + 1 U/µL RNAse inhibitor) until sorting. Quality assessment for each suspension includes fixation and paraffin embedding of cell pellets that are subsequently cut at 3 um using a rotary microtome and stained with H&E and cell markers, such as NeuN (ab177487, Abcam, Boston, MA, USA), MAP2 (ab183830, Abcam, Boston, MA, USA), and neurofilament (ab8135, Abcam, Boston, MA, USA); astrocyte marker GFAP (MAB360, Millipore, Burlington, MA, USA and Z0334, Dako, Santa Clara, CA, USA); and microglia markers IBA1 (-19741, Wako Richmond, VA, USA) and LN3 (ab166777, Abcam, Boston, MA, USA), as shown in Serrano et al. 2020 [55]. Adjacent paraffin source blocks were cut at 6 μm and stained with astrocyte marker GFAP (MAB360, Millipore, Burlington, MA, USA) to address astrocytes reactivity per group. Quantification was done using the percent of area occupied by Image J (version 1.48), and one-way ANOVA with Tukey’s Multiple Comparison Test was used to analyze group differences using GraphPad Prism Software (version 10.4).

### 4.3. Astrocyte Suspension Enrichment

Frozen cryoprotected single-cell suspensions were rapidly thawed and fixed with cold 70% Methanol with 2 mM EDTA for 30 min. Cells were magnetically sorted for GFAP using Dynabeads M-450 Epoxy (14011, Thermo Fisher Scientific, Carlsbad, CA, USA) and GFAP 488 antibody (194324, Abcam, Boston, MA, USA). RNA was isolated from the sorted cells using Qiagen Rneasy micro kit (#74004, Qiagen, Germantown, MD, USA) by adding 100 uL of RLT Lysis Buffer with B-Mercapoethanol to the bead-bound cells and then sonicating with light pulses (15 amps), placed in a magnetic rack for 1 min and removing the supernatant into a new microcentrifuge tube and proceeding with RNA Isolation following manufacturer’s instructions. RNA yield and RNA Integrity (RIN) were calculated on RNA 6000 picochip (5067-1513, Agilent, Santa Clara, USA) using Agilent 2100 Bioanalyzer. The library was sequenced by 2 × 150 bp paired-end sequencing on an Illumina NovaSeq S4. For a subset of cases, q-RT-PCR with pre-amplification was performed as described in Serrano et al. 2020 to determine enrichment for astrocytes [55]. TaqMan Gene Expression assays (4351370, Thermo Fisher Scientific, Carlsbad, CA, USA) included GFAP (Hs00909233_m1), MAP2 (Hs00258900_m1), and IBA1 (Hs00610419_g1), in addition to housekeeping probe ACTB (Hs01060665_g1), and were used to determine relative gene expression of astrocytes (GFAP), neurons (MAP2), and microglia (IBA1). RNA from cells positively sorted for GFAP was compared to RNA from the remaining cells not bound to the GFAP-magnified beads. In addition, we confirmed cell enrichment doing fGSEA plots using the plotEnrichmnet function described in Piras et al. (2021) [56]. In short, cell-specific gene set enrichment analysis was done using a list of 5641 cell-specific genes generated from a brain single-nucleus RNA sequencing dataset from the DLPFC [57]. Genes from all cases were pulled together, and the analysis was conducted using the R version 4.2.1 function enrichment package (fgsea, version 1.32.4), adjusting for multiple testing with FDR [56]. The analysis used pre-ranked gene lists, where genes were ranked based on log2FoldChange values.

### 4.4. Differential Expression Analysis

Sequencing reads were aligned to the Human Reference Genome GRCh38 using STAR [58] and summarized at the gene level using the HTSeq tool version 2.0.5 [58,59]. Quality controls were conducted using MultiQC v1.26 [60] software. Outlier detection was conducted through Principal Component Analysis (PCA), using R software v3.3.1. Samples with a total number of reads < 40 M and uniquely mapped reads < 55% were excluded from the downstream analysis. Genes with a total count of less than 10 were excluded. Normalization was performed using DEseq2. Gene expression differential analyses for every possible comparison between PSP, PD, ADD, ILBD, and control cases were performed using the R package DESeq2 v1.38.3 [61], including age, gender, and PMI as covariates. Significance was determined using Benjamini–Hochberg False Discovery Rate (FDR) correction, with genes meeting an adjusted *p*-value threshold of <0.05 considered differentially expressed.

In addition, to gain insight into the biological functions of the DEGs, we performed a gene set enrichment analysis (GSEA) using the cluster Profiler v4.6.2package in R software 4.4.3 [62]. The normalized enrichment score (NES) was computed to adjust for gene set size, and multiple testing correction was applied using the False Discovery Rate (FDR) method to control for false positives. Gene sets with an FDR-adjusted *p*-value < 0.05 were considered statistically significant.

### 4.5. Hub Gene Identification

In order to identify astrocyte-related Hub genes commonly affected in PSP, PD, and ADD, we used a search tool for the Retrieval of Interacting Genes (STRING) [63] to predict protein–protein interactions (PPI) of selected genes that were commonly dysregulated in all diseased cases when compared to CN. Cytoscape [64] was used to generate a network model from all genes with a confidence score ≥ 0.4 in STRING, by incorporating both physical and functional protein-protein interactions while CytoHubba, was used to identify Hub genes based on the connectivity of degree, DMNC, MCC, MNC, and closeness in the PPI network.

## Figures and Tables

**Figure 1 ijms-26-03416-f001:**
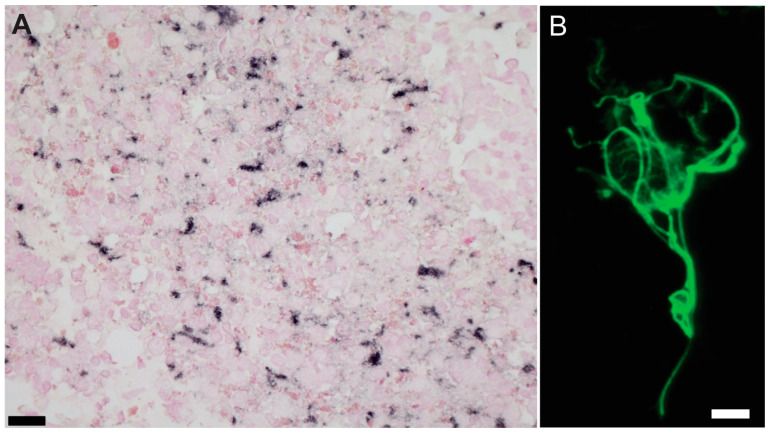
GFAP immunostains for quality assessments. (**A**) Whole-cell-dissociated-suspensions pellets were stained for GFAP before sorting to do quality assessments (Scale bar = 20 µm). and (**B**) after sorting to confirm astrocyte enrichment in our preparations. Example of whole astrocyte in suspension. (Scale bar = 10 µm).

**Figure 2 ijms-26-03416-f002:**
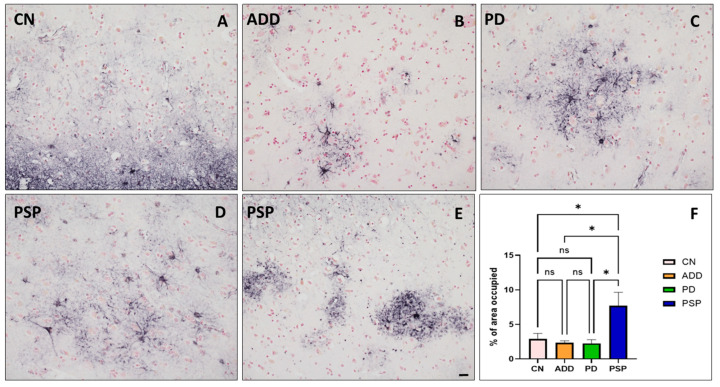
Frontal cortex tissue adjacent to the sample collected for whole-cell-dissociated-suspensions stained for GFAP showed an increased number of reactive astrocytes in PSP cases (**D**,**E**), especially when compared to (**A**) controls, (**B**) ADD, and (**C**) PD. (Scale bar on (**A**–**E**) = 50 µm). (**F**) The area occupied, measured using Image J version 1.48, showed a significant increase of GFAP positive reactive astrocytes compared to all other diagnostic groups (* *p* < 0.05 Tukey’s multiple comparisons test; ns = non statistically significant).

**Figure 3 ijms-26-03416-f003:**
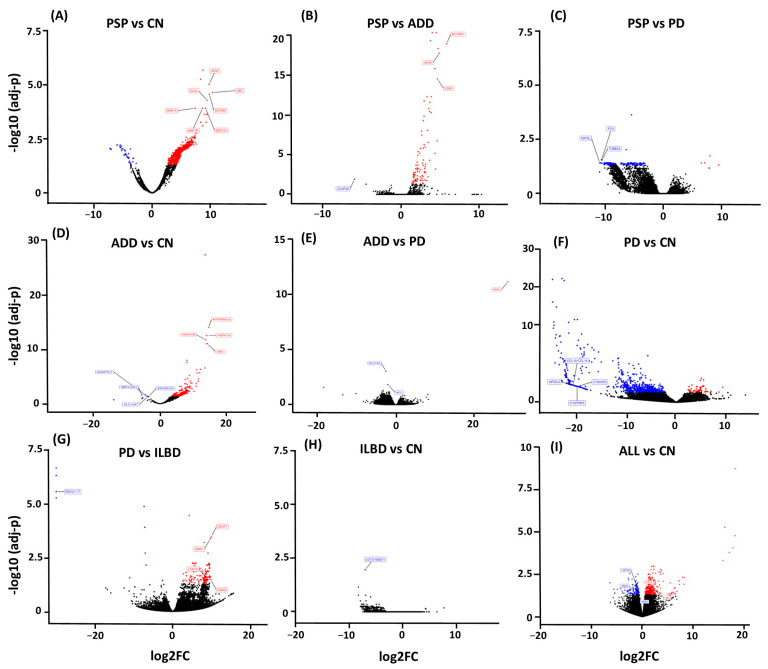
Volcano Plots show dysregulated genes obtained from differential expression analysis comparing gene expression from each diseased group to CN, as well as disease comparison. (**A**) PSP cases had the largest number of dysregulated genes when compared to CN with a total of 1933 upregulated genes (red) and 35 downregulated genes (blue). (**B**) When PSP was compared to ADD, we observed a total of 89 genes dysregulated. (**C**) and when compared to PD 243 genes. (**D**) ADD comparison to CN resulted in 754 genes dysregulated. (**E**) and only 4 genes when compared to PD. (**F**) While the PD comparison to CN resulted in 1150 dysregulated genes. (**G**) 135 when PD was compared to ILBD. (**H**) ILBD only had one gene dysregulated when compared to CN, and (**I**) all disease cases showed a total of 501 genes when all cases were combined and compared to CN.

**Figure 4 ijms-26-03416-f004:**
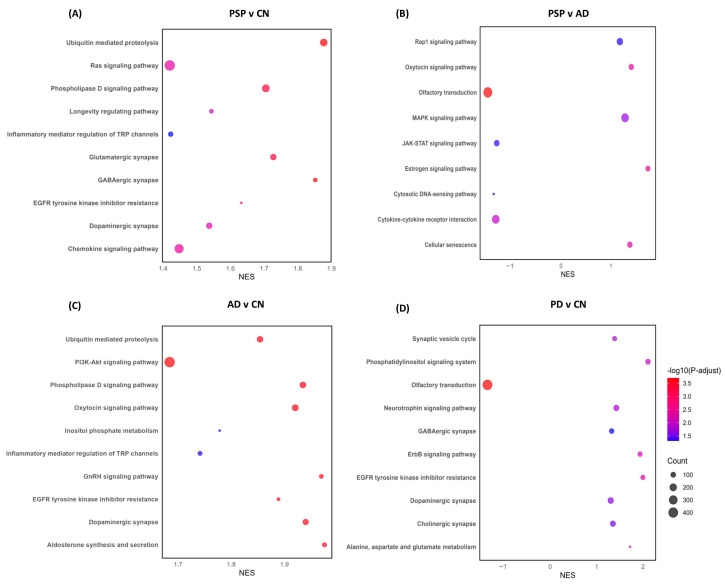
Pathway enrichment analysis was obtained from differential expression analysis comparing gene expression from each diseased group to CN, as well as disease comparison. Normalized enrichment score (NES) of genes associated with specific pathways (X axis) either downregulated (negative values) or upregulated (positive values). The size of each bubble correlates to the number of reads related to each pathway, with larger bubbles representing a greater number of genes, while the color correlates to greater or more statistically significant dysregulation, with red representing a stronger dysregulation than blue. (**A**) PSP and CN cases comparison showed an enrichment of dysregulated genes associated with multiple biological pathways, some of the most important involving synaptic pathways, chemokine signaling, protein modification processes and proteolysis, while compared to AD, we also observed dysregulation of cellular senescence, cytokine-cytokine receptor, Inositol phosphate metabolism, and other cellular processes and interactions (**B**). (**C**) ADD shows dysregulation of synapses, proteolysis, and signaling pathways when compared to CN. (**D**) Compared to CN, PD showed dysregulation of the translocation of olfactory receptors, synapses, synaptic vesicle cycle, and neurotrophic signaling pathways.

**Figure 5 ijms-26-03416-f005:**
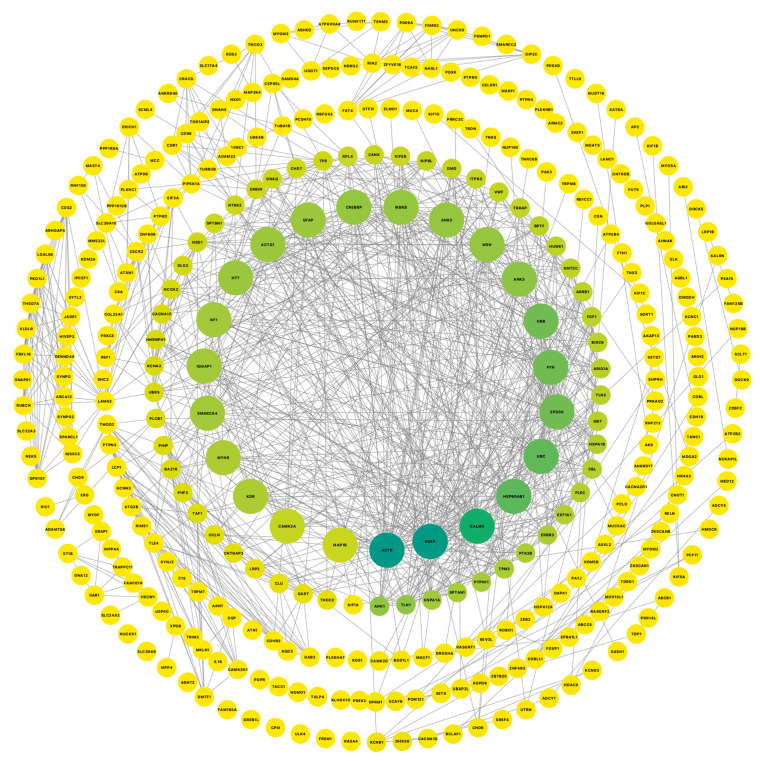
Cytoscape was used to visualize gene connectivity. Highly connected genes appear larger and green, while less connected genes are shown smaller and in yellow, indicating fewer interactions. The protein–protein interactions network from a list of 691 common dysregulated genes for each disease vs. CN comparison shows 541 nodes and 4952 edges; the overlapping analysis identified five common hub genes (dark green; ACTB, EGFR, CALM3, HSP90AB1, and UBC).

**Figure 6 ijms-26-03416-f006:**
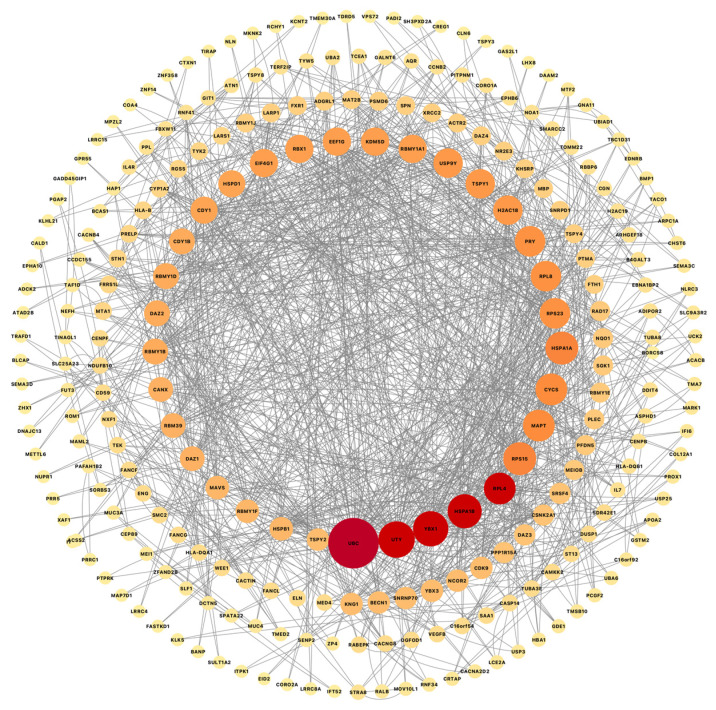
Cytoscape was used to visualize gene connectivity. Highly connected genes appear larger and red, while less connected genes are shown smaller and light orange, indicating fewer interactions. Protein–protein interactions network from a list of 501 common genes from a differential expression analysis combining all disease cases and comparing them to all non-symptomatic cases shows 268 nodes and 1124 edges, while the overlapping analysis identified five common hub genes (red; UBC, YBX1, UTY, RPL4, and HSPA1B).

**Figure 7 ijms-26-03416-f007:**
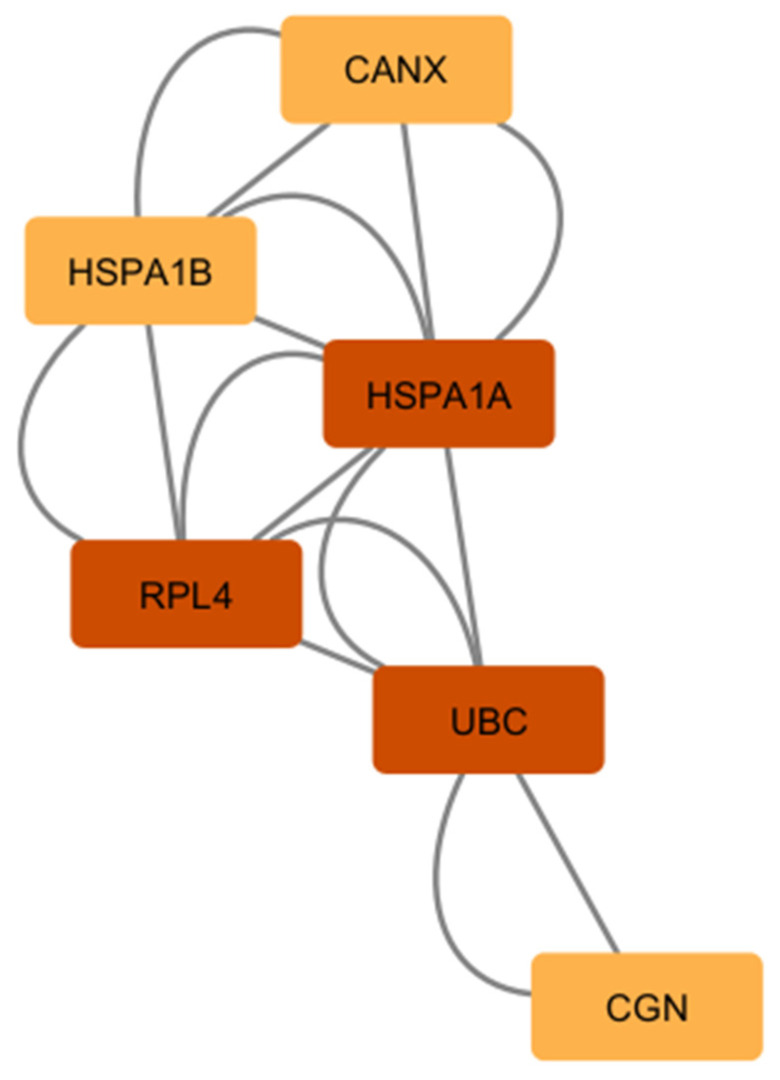
The list of common hub genes affected in astrocytes from multiple neurodegenerative diseases. We used the network and hub genes identified in Figure 5 and Figure 6 and identified and validated UBC, RPL4, and HSPA1A as important hub genes whose networks seem to be commonly affected in PSP, ADD, and PD astrocytes.

**Table 1 ijms-26-03416-t001:** Demographics and pathological summary. ^#^
*p* < 0.05 for group comparisons; * *p* < 0.05 for post-test comparison. (Kruskal-Wallis with Dunn’s Multiple Comparison Test); n: number of cases; SD: standard deviations; PMI, postmortem interval in hours; LB: Lewy bodies; CN: cognitively unimpaired low pathology controls; ILBD: incidental Lewy body; PSP: progressive supranuclear palsy; PD: Parkinson’s disease; ADD: Alzheimer’s disease dementia; RIN: RNA Integrity Number.

Diagnosis (n)	Age (SD)	PMI (SD)	LB Total (SD) ^#^	Tau Total (SD) ^#^	Plaque Total (SD) ^#^	PSP & Tau Pathology (SD) ^#^	RIN (SD)
CN (3)	89 (10)	2.9 (0.3)	0	6 (.6)	7 (6)	6 (0.6)	9.1 (0.2)
ILBD (3)	86 (9)	3.1 (0.7)	13 (2)	5 (3)	0.6(1)	5 (3)	9.5 (0.1)
PSP (5)	89 (7)	4.4 (1.1)	2 (4)	9 (1)	7 (6)	27 (9) *	8.6 (0.7)
PD (6)	82 (14)	3.7 (0.6)	31 (5) *	5 (1)	0.5 (1)	5 (1)	9.2 (0.4)
ADD (4)	87 (4)	3.2 (1.1)	0	10 (3) *	12 (3) *	10 (3)	9.0 (0.3)

**Table 2 ijms-26-03416-t002:** Differential gene expression found from each comparison. CN: cognitively unimpaired low pathology controls; ILBD: incidental Lewy body; PSP: progressive supranuclear palsy; PD: Parkinson’s disease; ADD: Alzheimer’s disease dementia.

Comparison (Number of Samples)	No. of DEGs
PSP (5) vs. CN(3)	1968 (1933 upregulated, 35 downregulated)
PSP (5) vs. ADD (4)	89 (88 upregulated, 1 downregulated)
PSP (5) vs. PD (6)	243 (4 upregulated, 239 downregulated)
ADD (4) vs. CN (3)	754 (749 upregulated, 5 downregulated)
ADD (4) vs. PD (6)	4 (1 upregulated 3 downregulated)
PD (6) vs. CN (3)	1150 (53 upregulated, 1097 downregulated)
PD (6) vs. ILBD (3)	150 (135 upregulated, 15 downregulated)
ILBD (3) vs. CN (3)	1 (downregulated)
PSP + PD + ADD (18) vs. CN + ILBD (6)	503 (364 upregulated, 139 downregulated)

## Data Availability

Data contained within the article will be posted in Synapse (Project ID: syn64618348).

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
