# Peer review of "Characterization of Isolated Human Astrocytes from Aging Brain"

_ijms, 2025, doi:10.3390/ijms26073416_

Round 1

Reviewer 1 Report

Comments and Suggestions for Authors

In this study, the authors aimed to characterize transcriptomic changes in human astrocytes isolated form postmortem brain tissue of individual with different neurodegenerative diseases. Using human postmortem brain samples significantly enhances the value of this study by avoiding the limitations of animal models or cell cultures. The research design of this study is appropriate and effectively addressing the study’s objectives. Comparing different type of neurodegenerative diseases allows identification of both common and disease specific conditions of astrocyte transcriptome. While the overall content of the work is valuable, the manuscript can be improved by providing more detailed explanations of key concepts, clarifying the presentation of the data and providing more data to validate accuracy. Please see the attached file. 

Author Response

In this study, the authors aimed to characterize transcriptomic changes in human astrocytes isolated form postmortem brain tissue of individual with different neurodegenerative diseases. The use of human postmortem brain samples significantly enhances the value of this study by avoiding the limitations of animal models or cell cultures and providing clinical relevance. The research design of this study is appropriate and well-structured, effectively addressing the study's objectives. Comparing different type of neurodegenerative diseases allows identification of both common and disease specific conditions of astrocyte transcriptome. While the overall content of the work is valuable, the manuscript can be improved by providing more detailed explanations of key concepts, clarifying the presentation of the data and providing more data to validate accuracy.

In principle, all figure legends should clearly describe what the figures represents. For example, figure legend 2 should include magnification levels, quantification method and statistics. They have the scale bar in the image but there is no indication of what the scale bar represents. It seems like figure 2F has higher magnification than the other images. If so, the authors should indicate that in the figure legend. The quantification should also be explained in the methods section. Is this a representation of their overall quantification? What is the percentage of the GFAP+ astrocytes for each group?

We thank the reviewer for their overall review and for this observation. We added more information in all the suggested legends, including the size of the scale bars in all histology images (fig1-2). 

We originally did not perform GFAP quantification of the frontal block adjacent to the samples used for astrocyte isolation because the images were intended to be a representation of reactive astrocytes in each disease group. However, to supplement this information, we now included quantification of a subset of cases used for the study. This is now also explained in the manuscript and legend. Panel F had a larger magnification to show a closer look of reactive astrocytes, but we removed it now to show the quantification graph.

Figure 1- It is not clear what figure 1B represents. It is a GFAP+ cell from the suspension? They mentioned in the methods section that they stained the cells with different cell markers and cited figure 1. However, we do not see any indication of neuronal, microglial or astroglia staining in the figure 1. If they did this experiment to assess the astrocyte enrichment and purity, this data should be included at least as a supplementary.

Figure 1 is the figure referring to observational quality assessments of astrocytes before and after enrichment. We modified the legend to make it clearer. During the editing process, the figure was moved to methodology and placed in the incorrect area which made it more confusing. Quality assessment of all the types of cells was already published on Serrano et al 2021 and for that reason we did not include an image on this article. We quantified a subset of cases using qPCR to ensure enrichment capabilities of our protocol and used Fast Gene Set Enrichment Analysis (FGSEA) to determine whether set genes from our sequencing data were significantly enriched.  We have now added a supplemental figure and more information about these methods.

The authors should report the purity level of the astrocytes post-sorting.

Added now more information about this in methods and results.

The authors should clarify if there is batch effect, and how it was handled in the transcriptomic analysis.

We sent all the data in one batch and therefore did not include batch effect in the analysis. This was originally written in the manuscript as a mistake but has now been removed.

The authors should include the statistical analysis in the methods.

Added in section 4.4

The authors should clarify what statistical tests were used for the analysis for table 1.

Added now in the methodology section (4.1) where we mentioned the demographic table and in the table description.

The color scale and size scale in the pathway analysis should be explained. What do they represent, enrichment score, fold change, or statistical significance? This should be clarified.

 Added now in the figure legend.

Figure 7. List of common hub genes affected in astrocytes from multiple neurodegenerative dis- eases. We use the network and hub genes identified in figures 8 and 9 and identified and validated UBC, RPL4, and HSPAlA as important hub genes which networks seems to be commonly affected in PSP, ADD, and PD astrocytes. Did they mean figures 6 and 7?

Thanks for pointing out that typo, it should say figure 5 and 6. We corrected the legend.

If possible, validating some key finding from transcription data would strengthen the data. Especially, it will ensure the clinical relevance.

We agree but due to the strict deadline and budgetary restriction we will not be able to do such experiments right now. We added now language expressing the desire to do future experiments for validation.

Discussion can be improved providing specific comparison of their findings with the previous studies. For example, one of their key findings of ubiquitin proteasome system. They can emphasize how UPS play role in neurodegeneration and what their changes suggests for mechanism.

We thank the reviewer for this comment. We added now more on this topic and cited more studies using similar disease groups.

For years scientists have noticed the presence of ubiquitin inclusions in multiple intracellular abnormally aggregated proteins that are pathological, such as tau and LB. This ap-pears to be a common feature for many disorders where proteins get abnormally aggregated intracellularly and seems to correlate with the common neurodegeneration observed in many neurological diseases. [47-50]. It has also been described how important the ubiquitin system is in glia cells, but to our knowledge none have shown dysregulation of this system in astrocytes from common neurodegenerative diseases [51]. Ubiquitination can alter the molecular functions of tagged substrates with respect to protein turnover, biological activity, subcellular localization, or protein–protein interaction. The UPS has been im-plicated in pathways that regulate neurotransmitter release, synaptic membrane receptor turnover and synaptic plasticity and as a result, affects a wide variety of cellular process-es, with chronic overexpression potentially inducing synaptic dysfunction in neurons.

Line 249-251 when comparing PD with control, there were down regulations of many micro RNBNA as well as chemokines and immunoglobins genes, suggesting of infflmmatory reposn in this group which has been previously suggested by others. Citation missing.

We have now added citations and improved the discussion by briefly explaining that this is not the most common finding from other studies.

When comparing PD with control, there were down regulations of many microRNA [29-31] as well as chemokines such as CXC14 and CXC15, and immunoglobins genes. This suggests a suppression of inflammatory response in this group which has been pre-viously suggested by others [29-33] but is not commonly observed in brain regions with higher LB-associated neurodegeneration, such as substantia nigra [34]. Our data seems to indicate that astrocytes in the frontal cortex of PD subjects, even though affected by LB pa-thology, do not have the same type of inflammatory response as other brain regions. A possible explanation is offered by Giovannoni, F. and F.J. Quintana, 2020 that proliferation of astrocytes and their reactivity varies depending on the type of injury and degree of damage in each brain region [35].  

Reviewer 2 Report

Comments and Suggestions for Authors

In FIg.1 authors describe  "an increased number of astrocytes", but is not present an image of the control.

Fig.1 is subsequent to Fig.2

In Fig.2 is not clear the comparison between images and is not  described what are the differences between  the  PSP cases shown;  is not present the magnification of the images.

In this study authors don't discuss about A1 astrocytes (neurotoxic phenotype) and A2 astrocytes (neurotrophic and neuroprotective).

Recently, Giovannoni and Quintana in Trend in immunology Vol 41 (2020)  have shown that "astrocyte proliferation is not a universal response to all damage, but is instead quite limited in contexts of infammation and neurodegeneration".... It would be important to discuss this aspect.

Authors don't discuss the role of the microglia but, as can be read in literature, the change of a quiescent astrocyte to a pro-infammatory phenotype is often mediated by microglia.

Author Response

In FIg.1 authors describe an increased number of astrocytes", but is not present an image of the control.

We thank the reviewer for their overall review and suggested changes that we believe improve our manuscript. We corrected the legend that was misleading, explained that the image represents quality assessment before and after enrichment and now added a supplemental figure describing further analysis done for enrichment analysis.

Fig.1 is subsequent to Fig.2

During the editing process, the figure was moved to methodology and placed in the incorrect area. We have now placed it in the results section where we had it originally. 

In Fig.2 is not clear the comparison between images and is not described what are the differences between the  PSP cases shown;  is not present the magnification of the images.

We originally did not perform quantification of the adjacent frontal blocked used for astrocyte isolation because the images were intended to be a representation of reactive astrocyte in each disease group. However, to supplement this information, we have now included quantification of a subset of cases used for the study. This is now also explained in the manuscript and legend. Panel F had a larger magnification to show a closer look of reactive astrocytes, but we removed it to show the quantification graph. We added scale bars to all pictures; this is more ideal in histology than reporting the microscopic magnification that will not consider when images get cropped and enlarged during editing. 

In this study authors don't discuss about A1 astrocytes (neurotoxic phenotype) and A2 astrocytes (neurotrophic and neuroprotective).

Most of the previous work on this is from animal and cell models. We looked at the limed genes listed by Liddelow, et al 2017 and we found that even though many were present in our sequencing data none of those genes were dysregulated in any of the diseased human cases when compared to aged controls except for GFAP (PAN astrocytes). We added a short statement to the discussion on neurotoxic vs neuroprotective role, but do not think it will be appropriate to expand on the classification when those genes previously published are not dysregulated in our samples as previously observed also by Liddelow, et al 2017.

Recently, Giovannoni and Quintana in Trend in immunology Vol 41 (2020)  have shown that "astrocyte proliferation is not a universal response to all damage, but is instead quite limited in contexts of infammation and neurodegeneration".... It would be important to discuss this aspect.

We also added a comment and cited Giovannoni and Quintana’s article when addressing the downregulation of some inflammatory markers observed in PD. We agree that the response will not be universal, but we have to be careful in making strong conclusions of our results and compared those to some of the cited manuscripts because the vast majority of the current understanding on astrocytes is derived from animal and invitro models, not human brains collected at autopsy.

When comparing PD with control, there were down regulations of many microRNA [29-31] as well as chemokines such as CXC14 and CXC15, and immunoglobins genes. This suggests a suppression of inflammatory response in this group which has been previously suggested by others [29-33] but is not commonly observed in brain regions with higher LB-associated neurodegeneration, such as substantia nigra [34]. Our data seems to indicate that astrocytes in the frontal cortex of PD subjects, even though affected by LB pathology, do not have the same type of inflammatory response as other brain regions. A possible explanation is offered by Giovannoni, F. and F.J. Quintana, 2020 that proliferation of astrocytes and their reactivity varies depending on the type of injury and degree of damage in each brain region [35].  Authors don't discuss the role of the microglia but, as can be read in literature, the change of a quiescent astrocyte to a pro-inflammatory phenotype is often mediated by microglia.

The idea of the study was to determine if we observed unmasked changes in astrocytes, we added a small comment about astrocyte interaction with other cells specially microglia but did not elaborate much on it because most of the discussed changes were intracellular changes related to astrocytes clearance rather than strong inflammatory changes.

It is well known that astrocytes interact and are affected by many different cell types influencing inflammatory response, especially microglia [18]. Liddelow, et al 2017 [19] and others have described how heterogenous astrocytes are and how depending on the type of astrocyte, reactivity of these cells might have neurotoxic or neuroprotective responses. Our astrocytes enriched samples included both types of astrocytes, class A and B astrocytes [19], but in this analysis we found that previously reported genes specific to type A and B astrocytes were not dysregulated in disease when compared to aged control

Round 2

Reviewer 1 Report

Comments and Suggestions for Authors

I appreciate the authors' efforts in addressing my suggestions. The manuscript has been revised satisfactorily, and I am pleased to recommend it for publication. My only suggestion is to use the terminology ‘neurotoxic (A1)’ and ‘neuroprotective (A2)’ as defined by Liddelow et. al. instead of ‘Class A’ and ‘Class B’ astrocytes for greater clarity and consistency with existing literature.

Reviewer 2 Report

Comments and Suggestions for Authors

The authors have taken on board the comments and have integrated and improved image descriptione